# QFT: Quantized Full-parameter Tuning of LLMs with Affordable Resources

## Abstract

Large Language Models (LLMs) have showcased remarkable impacts across a wide spectrum of natural language processing tasks. Fine-tuning these pre-trained models on downstream datasets provides further significant performance gains, but this process has been challenging due to its extraordinary resource requirements. To this end, existing efforts focus on parameter-efficient fine-tuning, which, unfortunately, fail to capitalize on the powerful potential of full-parameter fine-tuning. In this work, we propose QFT, a novel Quantized Full-parameter Tuning framework for LLMs that enables memory-efficient fine-tuning without harming performance. Our framework incorporates two novel ideas: (i) we adopt the efficient Lion optimizer, which only keeps track of the momentum and has consistent update magnitudes for each parameter, an inherent advantage for robust quantization; and (ii) we quantize all model states and store them as integer values, and present a gradient flow and parameter update scheme for the quantized weights. As a result, QFT reduces the model state memory to 21% of the standard solution while achieving comparable performance, e.g., tuning a LLaMA-7B model requires only <30GB of memory, satisfied by a single A6000 GPU.

## 1 Introduction

Large Language Models (LLMs), with up to hundreds of billions of parameters, have left an indelible mark on the landscape of natural language processing tasks, showcasing their remarkable impacts across a diverse spectrum of applications and domains (Touvron et al., 2023a;b; Brown et al., 2020; Zhang et al., 2022). Fine-tuning these pre-trained models on downstream datasets enhances their ability to understand and perform specific tasks (Zhao et al., 2023). However, due to the enormous number of parameters, the fine-tuning process requires unprecedented resources.

Parameter-efficient fine-tuning, involving the tuning of only selected parameters, is deemed a practical choice for low-resource situations (Ding et al., 2022; Hu et al., 2021; Li & Liang, 2021). Regrettably, owing to the limited representational capacity of the smaller parameter set, the outcomes of this approach often fall short of expectations (Lv et al., 2023). Therefore, our emphasis is placed on full-parameter fine-tuning, with a keen interest in investigating memory optimization strategies to render it feasible on cost-effective resources.

We begin by examining the full spectrum of memory usage in full-parameter fine-tuning, which can be categorized into three components: model states, activation, and other temporary or unusable memory. Model states, which include the model parameters (weights), gradients, and optimizer states (such as momentum and variances in Adam (Kingma & Ba, 2015)), are mandatory to store and consequently consume the majority of the memory (Rajbhandari et al., 2020). For instance, when employing the standard fp32 training settings with the Adam optimizer, the memory allocation for model parameters, gradients, momentum, and variances amounts to 4 times the number of parameters. As a result, tuning a LLaMA-7B model necessitates a minimum of 100.4GB of RAM, which presents a formidable challenge given the limitations of current GPU capacities.

In this work, we are motivated to reduce the memory usage of all model states through quantized low-precision representations. First, instead of resorting to straightforward quantization, we expect an optimizer that simplifies the computation to replace Adam. Fortunately, the Lion optimizer (Chen et al., 2023) aligns almost perfectly with our expectations, as it only keeps track of the momentum and naturally eliminates the memory usage of the variances. And more importantly, its update has

the same magnitude for each parameter, thus mitigating potential imbalances or inaccuracies in weight updates introduced by limited representation precision. Afterwards, we develop lightweight yet accurate quantizers for each model state, notably the dense-and-sparse quantizer (Kim et al., 2023) for weight parameters, which are then stored in the quantized integer format. During computation, these quantized representations are dequantized on-the-fly into the floating-point format to dynamically perform high-precision arithmetic. Moreover, we present a novel gradient flow scheme for the quantized weights to ensure proper error propagation and parameter updates in training.

More specifically, our contribution can be summarized as follows:

- We propose QFT, a novel Quantized Full-parameter Tuning framework for LLMs, which leverages quantization to optimize memory usage in fine-tuning without sacrificing performance. QFT can be seamlessly integrated into mainstream LLM training tools with minor modifications to a few training units, and is well compatible with existing memory optimization methods.

- We analyze the simplicity and memory efficiency of the Lion optimizer and confidently recommend it as the best choice for quantized fine-tuning. On this basis, we proceed to quantize all model states into the integer format, with each quantizer striking a balance between training accuracy and throughput. We also present a gradient flow scheme for the quantized weights.

- We perform instruction tuning on the pre-trained LLaMA-2 models and extensively evaluate performance on various benchmarks. The results demonstrate that our QFT, with memory usage reduced to 21%, achieves comparable performance to standard floating-point training.

## 2 RELATED WORKS

**Efficient Optimizer**    The primary optimizers employed for training transformer models are the Adam family (Kingma & Ba, 2015; Loshchilov & Hutter, 2017). They maintain a rolling average of the previous gradients to promote stable convergence in training. However, their optimizer states (momentum and variances) imposes an extra memory overhead proportional to the number of model parameters, and this becomes a significant burden as LLMs' parameters increase. To overcome the memory challenges of model states, there are various memory-efficient schemes. LOMO (Lv et al., 2023) utilizes a vanilla SGD optimizer for training LLMs, which unfortunately fails to ensure training performance due to the slow convergence and weak stability of SGD (Li et al., 2023). Another imperfect solution is to utilize an Adafactor optimizer (Shazeer & Stern, 2018), which, despite storing only aggregated information, is also beset by instability issues. In this work, we adopt the Lion optimizer (Chen et al., 2023), relying on its advantage of only keeping track of the momentum but achieving comparable convergence to Adam. More importantly, thanks to the sign operation, its update has the same magnitude for each parameter, which gives it a great potential for robust quantization of gradients and optimizer states.

**Quantization for Memory Optimization**    Most existing quantization methods focus on inference efficiency (Gholami et al., 2022; Dong et al., 2019; 2020; Kim et al., 2023; Li et al., 2022a;b; Li & Gu, 2022; Jacob et al., 2018), and recently, quantization is also believed to have great potential for optimizing training efficiency. Note that this research line is different from traditional quantization-aware training (QAT) (Jacob et al., 2018; Liu et al., 2023). QAT inserts fake quantization nodes on weights and activations in training, where parameter arithmetic and storage retains the floating-point format, and thus training efficiency is not improved. As a comparison, quantization-based memory optimization methods, which attempt to utilize low-precision units to store parameters, can effectively reduce the memory budget in training, and thus have received increasing attention. Bitsandbytes (Dettmers et al., 2021) introduces a block-wise quantization method to compress the memory of optimizer states. QLoRA (Dettmers et al., 2023) uses quantized values to store frozen pre-training weights, keeping only the adapters in the floating-point format. In this work, we propose a novel memory-efficient full-parameter fine-tuning framework for LLMs, in which all model states are stored as quantized integer values, enabling comprehensive memory compression without sacrificing fine-tuning performance.

**Other Memory Optimization Methods**    Other prominent memory optimization methods include offloading (Huang et al., 2020; Wang et al., 2018; Peng et al., 2020) and gradient checkpointing (Chen et al., 2016; Kumar et al., 2019; Jain et al., 2020; Kirisame et al., 2020). Activation

offloading offloads activation to external memory (e.g., CPU memory). It is worth noting that offloading comes at the cost of transferring data to another storage, which can increase execution time. Gradient checkpointing is a technique that discards activations in the forward pass and recomputes them in the backward pass as needed. This approach involves a trade-off between memory usage and computation cost. In addition, there are also customized schemes proposed for training LLMs. LOMO (Lv et al., 2023) fuses the gradient computation and the parameter update in one step. This method can reduce the memory usage of gradient tensors to O(1); however, there is a potential caveat as it is incompatible with gradient accumulation for scaling batch sizes, limiting it to unstable training with small batch sizes. In contrast, our framework is orthogonal and well compatible with all the above methods.

## 3 METHODOLOGY

### 3.1 LION OPTIMIZER

In a recent exploration of algorithm discovery through program search for neural network training, a novel optimization algorithm, Lion (EvoLved Sign Momentum), was conceived (Chen et al., 2023). The method explores an expansive program space while implementing program selection and simplification strategies. Lion stands out due to its simplicity and memory-efficiency, only tracking momentum, differing from adaptive optimizers by employing a consistent magnitude update for each parameter using the sign operation. Comparative studies with established optimizers, like Adam (Kingma & Ba, 2015) and Adafactor (Shazeer & Stern, 2018), underscored Lion's efficacy, leading to superior results in various domains, from image classification to language modeling. Particularly notable, Lion boosts the accuracy of Vision Transformers (ViT) on ImageNet, decreases pre-training compute on JFT, and surpasses Adam in training diffusion models. However, its advantages grow with increased training batch sizes and necessitate a lower learning rate than Adam, given the larger update norm resulting from the sign function.

Designing quantized fine-tuning algorithms involves working with limited-precision representations of parameters, gradients and momentum. This can lead to several challenges, including increased sensitivity to noise, potential accumulation of rounding errors, and other precision-related issues. We find Lion more suitable for the task of quantized fine-tuning, due to the following reasons:

- **Simplicity:** Lion is simpler and more memory-efficient since it only keeps track of the momentum. This reduced complexity might be beneficial when dealing with quantized values, where added algorithmic intricacies can amplify quantization errors.
- **Consistent Update Magnitudes:** Unlike adaptive optimizers, Lion ensures that updates have the same magnitude for each parameter, which is determined through the sign operation. In a quantized setting, this consistency can mitigate potential imbalances or inaccuracies in weight updates introduced by limited precision.
- **Memory Efficiency:** Memory usage is a common concern in quantized neural networks, especially when deploying on edge devices with constrained memory. Lion's memory efficiency (only tracking momentum) makes it a potentially better fit for such quantized settings than optimizers like Adam, which track more state variables.

### 3.2 QUANTIZATION

The Lion optimizer simplifies the composition of model states, which consist only of model weights, gradients, and optimizer momentum, resulting in a 25% reduction in memory usage compared to the memory-intensive Adam optimizer. However, it is imperative to recognize that these model states are still retained in the original floating-point format, a characteristic that can introduce redundant representations and, consequently, contribute to memory inefficiency. In light of this consideration, quantization, which involves the use of reduced-precision formats such as INT8 to represent neural networks, emerges as a compelling avenue for further memory optimization.

The field of quantization methods primarily emphasizes improving model inference efficiency, with limited attention paid to reducing training overhead (Dettmers et al., 2021). Our approach stands out through a comprehensive training memory compression, which is accomplished by quantizing all model states within the Lion optimizer and storing them as integer values. This sets our approach

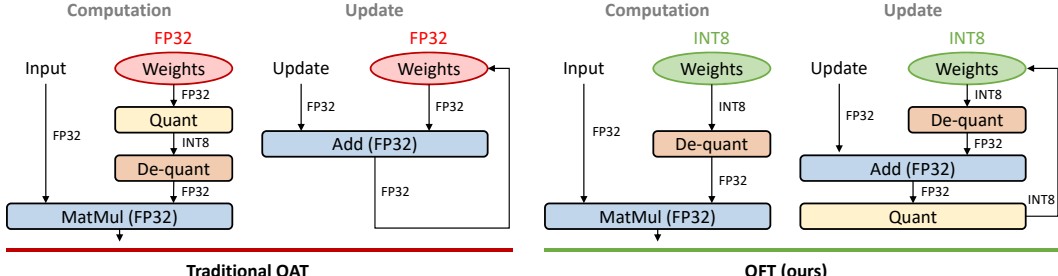

Figure 1: Comparison between our QFT and traditional QAT in the computation and update procedures of weights. QAT stores the weights in the floating-point format and adds fake quantization nodes to the computation. Conversely, in our QFT, the weights are stored in the low-precision integer format, which are de-quantized on-the-fly into the floating-point format for computation, resulting in a significant reduction in memory usage.

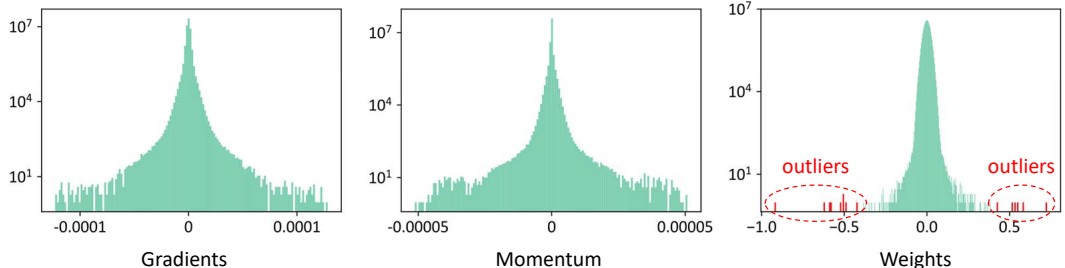

Figure 2: Illustration of the model state distributions when training a LLaMA-2-7B model. The weight values are from the final down projection layer, and the gradient and momentum values are fetched on the 200th training step. The gradients and momentum show a canonical centralized distribution with few outliers, while the range of the weights increases by three orders of magnitude and exhibits extreme outliers, posing a significant challenge to quantization.

apart from traditional QAT (Jacob et al., 2018). In our method, we initially store model parameters as quantized integers, whereas traditional QAT introduces fake quantization nodes to floating-point parameters. This distinction highlights the significance of our approach, as the latter method, with the reliance on fake quantization nodes, do not inherently enhance training efficiency. To more clearly demonstrate this difference, we present a comparison in Figure 1.

We first perform an in-depth examination of the numerical distributions of the model weights, gradients and optimizer momentum, as shown in Figure 2. This comprehensive analysis forms the basis for designing appropriate quantization strategies. Remarkably, we prioritize lightweight quantizers to minimize the impact of de-quantization on the training throughput. In the following, we describe in detail the quantizers employed for different model states.

**Uniform Quantizer for Gradients and Momentum** The gradients and momentum values exhibit a central distribution with few outliers that deviate from the central range, allowing us to confidently utilize the uniform quantizer, which is regarded as the most fundamental quantization method. The uniform quantizer includes two essential procedures: quantization and de-quantization, which are defined as follows:

$$Quant : \boldsymbol{X}^{(\mathbb{Z})} = \text{clip}\left(\left\lfloor \frac{\boldsymbol{X}}{s} \right\rceil + z, 0, 2^b - 1\right) \quad De\text{-}quant : \hat{\boldsymbol{X}} = s\left(\boldsymbol{X}^{(\mathbb{Z})} - z\right) \approx \boldsymbol{X} \quad (1)$$

where $\boldsymbol{X}$ is the floating-point vector, $\boldsymbol{X}^{(\mathbb{Z})}$ is the quantized integer vector, $\lfloor \cdot \rceil$ denotes the round function, and $b \in \mathbb{N}$ is the quantization bit-width. $s \in \mathbb{R}^+$ and $z \in \mathbb{Z}$ are the quantization scale and zero-point, respectively, and with fast computational considerations, they are directly determined by the arithmetic lower and upper bounds of $\boldsymbol{X}$ as follows:

$$s = \frac{\max(\boldsymbol{X}) - \min(\boldsymbol{X})}{2^b - 1}, \quad z = \left\lfloor -\frac{\min(\boldsymbol{X})}{s} \right\rceil \quad (2)$$

**Dense-and-Sparse Quantizer for Weights**   In contrast to gradients and momentum, whose probability distributions lend themselves well to quantization, the weights present a distinct challenge. This challenge arises from their considerably broader range, which is approximately three orders of magnitude larger than that of momentum, as well as the presence of pronounced outliers. This combination of factors makes the accurate quantization of weights a particularly formidable task (Kim et al., 2023; Frantar et al., 2022; Lin et al., 2023).

Upon revisiting the weight distribution, we uncover an intriguing pattern: if we set aside the extreme outliers, the remaining parameters coalesce into a notably compact distribution. To elucidate, the initial expansive range is predominantly influenced by these extreme outliers, with a striking statistic that 99% of the values cluster within a mere 20% of the overall range. This revelation serves as the catalyst for our approach, drawing inspiration from the dense-and-sparse quantizer presented in (Kim et al., 2023). This method effectively ameliorates the issue of outliers by decomposing the weights into two distinct matrices: one dense and the other sparse. Formally, the method is defined as follows:

$$\boldsymbol{W} = \boldsymbol{D} + \boldsymbol{S} \ \text{ s.t. } \ \boldsymbol{D} = \boldsymbol{W}[T_{\min} \leq w \leq T_{\max}]$$
$$\text{and } \ \boldsymbol{S} = \boldsymbol{W}[w < T_{\min} \ \text{ or } \ w > T_{\max}] \tag{3}$$

where $\boldsymbol{D}$ is a dense matrix representing the centralized values, and $\boldsymbol{S}$ is a sparse matrix representing the outliers. Here, $T_{\min}$ and $T_{\max}$ are the thresholds for identifying outliers, which can be determined by the percentage of the range. It's important to highlight that the matrix decomposition process is numerically straightforward, ensuring a high level of computational efficiency with minimal repercussions on training overhead.

Subsequently, the dense matrix adheres to the simple uniform quantizer as described in Equation 1, while the sparse matrix retains its data in the floating-point format. Notably, given that the outliers constitute a relatively minor fraction, such as 1%, the sparse matrix can capitalize on memory-efficient storage techniques, like compressed sparse row (CSR) format, which can be instrumental in substantially mitigating memory overhead.

### 3.3   OVERALL FRAMEWORK

In this section, we integrate the above efficient Lion optimizer and quantization methods and introduce a memory-efficient fine-tuning framework for LLMs. We provide a comprehensive description of each training phase, including forward propagation, backward propagation, and parameter update, with particular emphasis on the quantized gradient flow and the quantized optimizer step.

---

**Algorithm 1** Gradient Flow of Quantized Weights

\# $T_l$ : saved tensors in forward pass of layer $l$
\# $g_o$ : gradient of the current layer's output

$S_g \leftarrow \texttt{stack}()$
**for** $l = L, L-1, \cdots, 1$ **do**
   $I_l, W_l^{(\mathbb{Z})} \leftarrow T_l$
   $W_l \leftarrow \texttt{dequant}(W_l^{(\mathbb{Z})})$
   **calculate gradients of $I_l$ and $W_l$**
   $g_i \leftarrow \texttt{matmul}(g_o, W_l)$
   $g_w \leftarrow \texttt{matmul}(g_o^T, I_l)$
   $g_w^{(\mathbb{Z})} \leftarrow \texttt{quant}(g_w)$    ▷ store as INT8
   $\texttt{push}(S_g, g_w^{(\mathbb{Z})})$    ▷ collect gradient
   **assign $g_o$ of layer ($l$-1)**
   $g_o \leftarrow g_i$
**end for**

---

**Algorithm 2** Quantized Lion Optimizer

\# $\beta_1, \beta_2, \lambda, \eta, f$ : optimizer parameters
\# $m_l$ : optimizer momentum of layer $l$

**for** $l = 1, 2, \cdots, L$ **do**
   $g_w^{(\mathbb{Z})} \leftarrow \texttt{pop}(S_g)$    ▷ retrieve gradient
   $g_w \leftarrow \texttt{dequant}(g_w^{(\mathbb{Z})})$
   $m_l \leftarrow \texttt{dequant}(m_l^{(\mathbb{Z})})$
   $W_l \leftarrow \texttt{dequant}(W_l^{(\mathbb{Z})})$
   **update model parameters**
   $\Delta \leftarrow \beta_1 m_l + (1 - \beta_1) g_w$
   $W_l \leftarrow W_l - \eta(\texttt{sign}(\Delta) + \lambda W_l)$
   **update EMA of $g_w$**
   $m_l \leftarrow \beta_2 m_l + (1 - \beta_2) g_w$
   $m_l^{(\mathbb{Z})} \leftarrow \texttt{quant}(m_l)$    ▷ store as INT8
   $W_l^{(\mathbb{Z})} \leftarrow \texttt{quant}(W_l)$    ▷ store as INT8
**end for**

**Quantized Forward Propagation**   Within our framework, we initially represent weights as quantized integer values to optimize memory utilization. During the execution of forward propagation, we de-quantize these low-precision weights into the floating-point format on-the-fly, thereby enabling high-precision arithmetic operations. For more clarity, we visualize this critical process in Figure 1.

**Quantized Backward Propagation**   In the backward propagation phase, the final task loss is propagated forward from the last layer in a sequential manner, and throughout this process, the gradient of each parameter is computed. It's worth noting that these gradients need to be kept in memory, as they serve as essential information for guiding subsequent updates to the parameters. However, in mainstream deep learning frameworks like PyTorch, only parameters in the floating-point format can possess the gradient property, while those in the integer format cannot. Consequently, we cannot compute and store the gradients using the automatic differentiation functionality (i.e., AutoGrad) in such cases. To this end, we design the gradient flow of integer weights, as presented in Algorithm 1. As in forward propagation, we begin by de-quantizing the weights into the floating-point format. Subsequently, leveraging the gradient of the output, we apply the chain rule to compute the gradients of both the input and the weights. Beyond the computational aspect, preserving the gradients of the weights presents its own set of formidable challenges. To address this, we introduce a gradient retention scheme centered around the maintenance of a global stack. In this scheme, the gradient of each layer is sequentially pushed to the stack, following the backward flow of information during the backward propagation.

**Quantized Parameter Update**   Ultimately, the parameter update are executed in accordance with the Lion optimizer procedures, with the notable difference that the gradients and momentum are stored in the integer format. The quantized optimizer step is outlined in Algorithm 2. Initially, we pop the elements from the global stack to access and retrieve the gradients. It is essential to emphasize the exceptional computational efficiency of this popping process, as its computational complexity consistently remains at O(1), independent of the stack length. This efficiency arises from a distinct pattern: in the backward propagation phase, the gradients are sequentially pushed into the stack, beginning from the last layer. Conversely, in the optimizer step, the gradients are popped in a sequential manner, commencing from the first layer. This strategic arrangement ensures that the gradient of the current layer always occupies the last position in the stack, fully capitalizing on the first-in-last-out property inherent to stack data structures.

## 4 EXPERIMENTS

### 4.1 EXPERIMENTAL SETUP

**Models and Benchmarks**   We conduct adequate evaluation of the proposed QFT by fine-tuning the advanced pre-trained model, LLaMA-2 (Touvron et al., 2023b), including the 7b and 13b versions. The few-shot performance of fine-tuned models is comprehensively evaluated on a variety of standard benchmarks, including ARC (Clark et al., 2018), HellaSwag (Zellers et al., 2019), MMLU (Hendrycks et al., 2020), and TruthfulQA (Lin et al., 2021). All results are obtained using the Language Model Evaluation Harness tool (Gao et al., 2021). In addition, we also use MT-Bench (Zheng et al., 2023) with GPT-4 scores to evaluate the conversational abilities of the models.

**Dataset Preparation**   In our experiment, we utilized a dataset comprising 94.1K shareGPT entries (HuggingFace, 2023b; shareGPT, 2023), which encompass user interactions with chatGPT. We adopted the data cleaning procedures from Fastchat (Chiang et al., 2023), converting HTML to markdown, eliminating non-English conversations, and segmenting extended dialogues into sequences capped at a length of 2048.

**Baseline Methods**   We evaluate QFT in terms of both training memory and performance. For training memory, QFT is compared to floating-point Adam (Kingma & Ba, 2015), Lion (Chen et al., 2023), as well as bitsandbytes with quantized optimizer states (Dettmers et al., 2021). For the performance of instruction tuning, we take Vicuna (Chiang et al., 2023), which performs full-parameter fine-tuning in the floating-point format, as the baseline method. For a fair comparison, we reproduce its results using the same dataset as QFT.

Table 1: Memory usage (in GB) when fine-tuning the LLaMA-2-7b model using different methods. We report the full spectrum of memory profiles, as well as the total allocated memory and peak allocated memory. For model states, the Lion optimizer in floating-point format provides a 25% memory reduction, and further, our QFT introduces quantization that reduces the memory to 21% of the Adam optimizer, allowing for fine-tuning within 30GB of RAM.

| Method | Weights | Gradients | Optimizer States | | | Activation | Total | Peak |
| | | | Weight Copies | Momentum | Variances | | | |
|---|---|---|---|---|---|---|---|---|
| Adam | 25.1 | 25.1 | - | 25.1 | 25.1 | 3.75 | 104 | 129 |
| Adam-mixed | 12.6 | 12.6 | 25.1 | 25.1 | 25.1 | 3.75 | 104 | 123 |
| bitsandbytes | 12.6 | 12.6 | 25.1 | 6.31 | 6.31 | 3.75 | 66.6 | 86.6 |
| Lion | 25.1 | 25.1 | - | 25.1 | - | 3.75 | 79.1 | 101 |
| QFT | 7.42 | 7.06 | - | 7.06 | - | 3.75 | 25.3 | 28.9 |

**Training Details** During training, we apply channel-wise quantization for all quantizers of model states. The threshold $T$ in the dense-and-sparse quantizer is obtained from 1% of the distribution range (please see Appendix A.1 for details). The training parameters are set to align with Vicuna's settings: the global batch size is 128, the learning rate is 2e-5, and the total number of epochs is 3.

## 4.2 MEMORY PROFILE

We start by discussing the memory usage using different methods, and the results of fine-tuning the LLaMA-2-7b model are reported in Table 1. In the training that employs the Adam optimizer with standard settings, it becomes evident that the memory consumption becomes significantly substantial. Specifically, the model weights, gradients, momentum, and variances each occupy a considerable 25.1GB of RAM, which is 4 times the model parameters, resulting in a horrible resource burden. Remarkably, this memory issue persists when employing the Adam optimizer with mixed precision settings. Despite the fact that the numerical precision of both weights and gradients experiences a 50% reduction during the forward and backward computations, the necessity to uphold full-precision weight copies within the optimizer states remains paramount. This stringent requirement is essential to guarantee the stability of parameter updates, as discussed in detail in Appendix A.2, and thus the goal of conserving memory remains unattainable.

The Lion optimizer simplifies the optimizer states by only keeping track of the momentum, resulting in a noteworthy reduction in memory usage, 25% less than that of the Adam optimizer. Hence, it takes up 25% less memory than the Adam optimizer. Notably, the model states still retain the floating-point format, and this redundant representation offers additional opportunities for optimization. To this end, bitsandbytes employs quantization methods to convert the momentum and variances into the integer format, resulting in an impressive memory savings of 37 GB. Nevertheless, the retention of floating-point weights and gradients remains a hurdle, preventing complete memory conservation and continuing to strain the training resources.

Our QFT, built on top of the Lion optimizer, employs a comprehensive quantization scheme encompassing all model states, including weights, gradients, and optimizer momentum. These parameters can be efficiently stored in the low-precision integer format. This allows the GPU to allocate only 21.5GB of RAM to store these parameters, marking a remarkable reduction to a mere 21% in comparison to the memory requirements of the Adam optimizer. During the practical training process, when taking into account factors such as activation, as well as several caches and memory fragments, the peak allocated memory remains comfortably below 30GB, allowing us to fine-tune within budget-friendly computing resources.

## 4.3 PERFORMANCE EVALUATION

In this section, we conduct a comprehensive evaluation of the instruction fine-tuning performance in both conventional and advanced manners, which are in turn compared and analyzed in detail below. In addition, we also provide a qualitative analysis of the model's language generation capabilities in Appendix A.3.

Table 2: Few-shot performance of different models on various standard benchmarks. Here, the number of shots is aligned to Open LLM Leaderboard (HuggingFace, 2023a). We take the pre-trained LLaMA-2 model as the baseline and compare the instruction tuning results of our QFT and Vicuna. Our QFT, with less resource consumption, encouragingly provides substantial improvement over pre-trained models and rivals the outcomes of full-precision tuning.

| Model | ARC-c (25-shot) | HellaSwag (10-shot) | MMLU (5-shot) | TruthfulQA-mc (0-shot) | Average |
|---|---|---|---|---|---|
| LLaMA-2-7B | 53.1 | 78.6 | 46.9 | 38.8 | 54.4 |
| Vicuna-7B* | 53.6 | 77.3 | 49.4 | 51.5 | 58.0 |
| LLaMA-2-7B-QFT | 52.9 | 76.7 | 48.8 | 51.1 | 57.4 |
| LLaMA-2-13B | 59.4 | 82.1 | 55.8 | 37.4 | 58.7 |
| Vicuna-13B* | 57.0 | 81.2 | 55.8 | 50.9 | 61.2 |
| LLaMA-2-13B-QFT | 56.2 | 81.0 | 55.9 | 48.6 | 60.4 |

**Few-Shot Evaluation** We perform few-shot performance evaluations across a range of well-established benchmarks to assess the effectiveness of QFT. The obtained results, pertaining to various model configurations, are comprehensively presented in Table 2. To maintain consistency, we opt to employ the same evaluation metrics as those employed in Open LLM Leaderboard (HuggingFace, 2023a) and ensure alignment with key experimental settings, such as the number of shots. As we can see, when fine-tuning a LLaMA-2-7B model, it becomes evident that QFT introduces a remarkable enhancement in performance. Specifically, QFT substantially elevates the average performance score, catapulting it from an initial value of 54.4 to a significantly improved 57.4. Impressively, this achievement positions QFT within a mere 0.6 points of the Vicuna model, which has undergone full-precision tuning. Regarding specific individual metrics, such as 5-shot MMLU, we observe an improvement in results from 46.9 to 48.8, highlighting the model's enhanced problem-solving capability.

Furthermore, it is imperative to provide a clarification regarding the observed slight decline in the 10-shot HellaSwag results across both fine-tuning settings. This diminution can be attributed, in part, to the influence exerted by the fine-tuning dataset and, in part, to the inherent limitations of a single benchmark evaluation, which may introduce a certain degree of one-sidedness or even inaccuracies into the assessment process (Liao et al., 2021). Consequently, it becomes increasingly evident that the central focus should shift to a careful comparison between the performance of Vicuna and QFT rather than dwelling extensively on the improvement of the pre-trained model itself, and it is indeed reassuring to note that QFT consistently demonstrates the ability to achieve results comparable to those achieved by the Vicuna model.

**MT-Bench Score** Besides the conventional benchmarks described above, there is a more advanced benchmark, MT-Bench, to evaluate the conversational abilities of LLMs. MT-bench consists of a series of challenging multi-round open-ended questions that match the characteristics and preferences of human conversations, and uses GPT-4 as a judge to automatically score the responses. The score results are reported in Table 3. As an illustrative example, we provide a detailed discussion of the 7B models. Initially, the LLaMA-2 model, in its pre-trained state, yields a rather modest score of 3.83, indicating a considerable limitation in its problem-solving ability. For the Vicuna model tuned in full precision, the score undergoes a substantial augmentation, surg-

Table 3: MT-Bench scores using GPT-4 of different models. They can reflect the conversational abilities of these models. Our QFT significantly outperforms the pre-trained LLaMA-2 model, and achieves comparable results to the Vicuna model tuned in full precision.

| Model | MT-Bench Score (GPT-4) |
|---|---|
| GPT-3.5 | 7.94 |
| LLaMA-2-7B | 3.83 |
| Vicuna-7B* | 6.08 |
| LLaMA-2-7B-QFT | 5.95 |
| LLaMA-2-13B | 4.69 |
| Vicuna-13B* | 6.46 |
| LLaMA-2-13B-QFT | 6.27 |

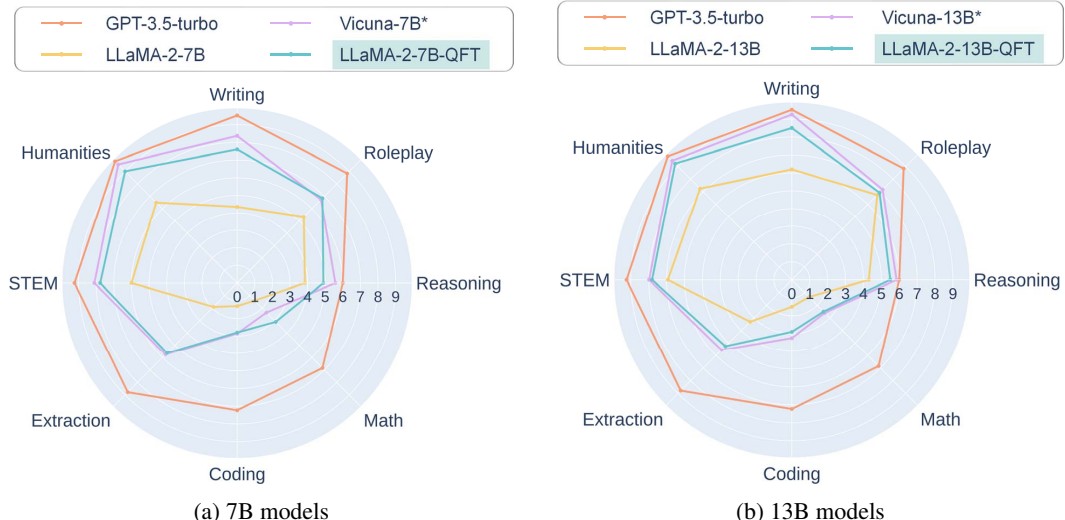

(a) 7B models          (b) 13B models

Figure 3: Radar charts of each capability in MT-Bench of different models. Compared to the pre-trained LLaMA-2 model, our QFT yields across-the-board improvements in all metrics. Compared to the Vicuna model tuned in full precision, our QFT achieves similar results and even surpasses it in some abilities, such as the Math metrics in the 7B model setting.

ing to an impressive 6.08. Notably, the QFT also mirrors this impressive improvement, boosting model performance to levels comparable to the Vicuna model.

To facilitate a more visual comparison, we provide radar charts that encompass eight capacity indicators, as illustrated in Figure 3. These radar charts clearly shows that QFT provides a comprehensive and transformative improvement across all measured metrics compared to the baseline performance of the pre-trained LLaMA-2 model. In comparison to the Vicuna model tuned in full precision, QFT achieves comparable results and even outperforms it in certain aspects, e.g., in the 7B model setting, QFT exhibits superior performance in the Math metrics.

## 5  CONCLUSIONS AND BROADER IMPACTS

In this paper, we propose a Quantized Full-parameter Tuning (QFT) framework for LLMs, which leverages quantization techniques to comprehensively optimize training memory to enable fine-tuning on affordable resources. We employ the memory-efficient Lion optimizer, which provides significant advantages for robust quantized fine-tuning. Upon this, we develop customized quantizers to store all model states in the integer format, significantly reducing the memory usage. QFT incorporates these two innovations and designs a novel gradient flow scheme to accommodate them. We perform instruction tuning on the pre-trained LLaMA-2 models to verify the effectiveness of QFT, and the results demonstrate that QFT can reduce memory usage to 21% while achieving comparable performance to standard floating-point training.

QFT can be easily integrated into mainstream LLM training tools and offers great compatibility with other memory optimization methods, demonstrating remarkable adaptability and utility in real-world applications. Additionally, it has the potential to produce broader impacts:

- **Quantized Training from Scratch:** The parameters to be updated and optimizer configurations in the full-parameter tuning are consistent with the pre-training process, thus QFT can be migrated to be applied to training-from-scratch cases.

- **Lower-Precision Optimizer Momentum:** Recent research has explored the compression of optimizer states to 4-bits (Li et al., 2023). It holds promise to explore the combination of QFT with this approach for even more substantial memory reduction.

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

# A APPENDIX

## A.1 DISCUSSION ON OUTLIER THRESHOLDS OF WEIGHT QUANTIZER

In this section, we discuss the selection and updating strategies for outlier thresholds in dense-and-sparse quantizers. We first report the memory and accuracy of dense-and-sparse quantizers using different percentage thresholds, and the results are shown in Table 4. The accuracy, i.e., the degree of distributional approximation of the quantizers, is evaluated by $L_2$ distance between de-quantized weights $\hat{W}$ and full-precision weights $W$, where the quantized weights are from the final down projection layer.

Table 4: Comparison of memory (in GB) and accuracy of dense-and-sparse quantizers using different percentage thresholds for weights. Here, accuracy is measured by $L_2$ distance between de-quantized $\hat{W}$ and full-precision $W$.

| Percentile | 0 | 0.45% | 1.0% | 3.0% | 5.0% |
|---|---|---|---|---|---|
| Memory | 7.06 | 7.23 | 7.42 | 8.23 | 9.16 |
| $L_2$ Distance | 436 | 0.846 | 0.619 | 0.566 | 0.479 |

The benefits of employing matrix decomposition in dense-and-sparse quantizers are readily evident from the results. When the percentile is set to 0, the quantizer no longer filters out the outliers and degenerates into a standard uniform quantizer, resulting in intolerable quantization errors. A noteworthy value suggested in SqueezeLLM (Kim et al., 2023) is 0.45%. However, it's essential to acknowledge that this tight bound is primarily designed for inference scenarios. To accommodate potential fluctuations during training, some level of relaxation in the thresholds is necessary. To this end, we assess the relaxed constraints at percentiles of 1.0%, 3.0%, and 5.0%. The findings reveal that, although the 3% and 5% percentiles yield a slight boost in accuracy, they also incur higher memory usage. Consequently, we advocate for the 1% percentile as a more favorable choice for achieving a balance between memory conservation and accuracy.

We also explore the update frequency of the thresholds. In order to maintain a high training throughput, we adopt a strategy of lazy updates, meaning that the thresholds set in the first iteration are reused in subsequent iterations. This approach is viable due to the smooth numerical updates of the weights during the fine-tuning process across neighboring iterations. Additionally, to our surprise, we discover that the initial thresholds remain effective for an entire epoch. As a result, we only update the thresholds once at the beginning of each epoch.

## A.2 DISCUSSION ON TRAINING STABILITY

In this section, we delve into the essentiality of maintaining full-precision weight copies in mixed-precision training and provide a comprehensive illustration of the benefits of QFT through a comparative analysis. In mixed-precision training, both forward and backward propagation involve weights and gradients in the FP16 format. However, during parameter updates within the optimizer, the weights are involved in calculations using the FP32 format. This approach is employed to circumvent potential data overflow issues that may arise in the FP16 format (Micikevicius et al., 2017), given its limited representation range. Consequently, maintaining full-precision weight copies becomes indispensable to ensure the stability of the training process.

Conversely, QFT effectively addresses the aforementioned issue without necessitating extra storage for weight copies. Specifically, unlike the rudimentary numerical truncation encountered in the FP16 format, the quantization procedure in QFT maps FP32 format values as a cohesive unit to a set of evenly distributed integer values, as illustrated in Equation 1. Consequently, when performing the calculations, these integer values are dynamically de-quantized into the FP32 format, without any potential risk of data overflow.

### A.3 QUALITATIVE ANALYSIS OF CONVERSATIONAL ABILITIES

In the experimental section, we demonstrate the effectiveness of QFT through quantitative evaluations of LLMs' performance on a variety of benchmarks. Despite the large number of benchmarks, the results of the machine evaluations are not considered to be comprehensive, and in some cases, their reliability may be subject to scrutiny, e.g., the "shortcut" issue. Consequently, in the following, we intuitively show and compare examples generated by LLMs, aiming to further verify the conversational abilities through qualitative analysis.

**[Writing Ability]**

*Question*: Write a captivating movie review for a recently released science fiction film, discussing its plot, characters, and special effects.

**Vicuna-7B\***: "The Last Voyage" is a thrilling science fiction film that follows a group of astronauts on a mission to explore a distant planet. When they arrive, they discover that the planet is inhabited by a mysterious alien race. As they try to communicate with the aliens and gather data, they realize that their mission may be more dangerous than they ever imagined. With stunning special effects and a gripping plot, "The Last Voyage" is a must-see for science fiction fans.

**LLaMA-2-7B-QFT**: "Intergalactic Intrigue" is a thrilling science fiction film that takes viewers on a journey through the stars. The film follows a group of space explorers as they embark on a mission to uncover the secrets of an ancient alien civilization. The film's special effects are top-notch, with stunning visuals that transport viewers to a world of wonder and discovery. The characters are well-developed and likable, with each member of the crew bringing their own unique skills and personalities to the table.

**[Roleplay Ability]**

*Question*: As a superhero, how would you explain your origin story to a curious child?

**Vicuna-7B\***: I was just like any other kid, until one day I was exposed to a strange chemical that gave me incredible powers. I learned to control the elements and use them to protect the innocent. I became known as the Elemental Guardian, and I've been fighting crime and saving the day ever since.

**LLaMA-2-7B-QFT**: I was just like any other kid, playing in the park and having fun. But one day, I was hit by a strange ray that gave me superpowers. I could fly, turn invisible, and even control the weather! I decided to use my powers to help people and fight crime. That's how I became the superhero you know and love!

**[Common-sense Ability]**

*Question*: In a world where automation is becoming increasingly prevalent, is it more important to prioritize job creation or technological progress?

**Vicuna-7B\***: Job creation is essential for a thriving economy, as it provides individuals with employment opportunities and a source of income. While technological progress is important for driving innovation and improving efficiency, it is crucial to balance technological advancements with the creation of new jobs. This can be achieved by investing in education and training programs that help workers develop the skills necessary to adapt to new technologies and by promoting policies that support small businesses and entrepreneurship. By prioritizing both job creation and technological progress, we can ensure a more stable and prosperous future for all.

**LLaMA-2-7B-QFT**: In the short term, job creation is a critical concern for policymakers and businesses. The unemployment rate remains high, and many people are struggling to find work. However, in the long term, technological progress is essential for driving economic growth and improving the standard of living for everyone. Automation has the potential to increase productivity and efficiency, which can lead to higher wages and better working conditions for employees. However, it can also lead to job displacement, particularly in industries where automation is most prevalent.

**[Fermi Ability]**

*Question*: How many snowflakes fall during a typical winter? Try to explain your answer. Your explanation should take the reader through your reasoning step-by-step.

**Vicuna-7B\***: It is difficult to determine the average number of snowflakes that fall during a typical winter, as it can vary greatly depending on location, climate, and other factors. However, it is estimated that on average, a single snowflake can have up to 100,000 ice crystals per flake. Over the course of a typical winter, this could result in billions of snowflakes falling from the sky.

**LLaMA-2-7B-QFT**: To answer this question, we need to consider the average snowfall for a typical winter in a given location. In the United States, for example, snowfall tends to be heaviest in the northern states, where the average snowfall for a typical winter is around 60 inches. In the southern states, snowfall is much less common and the average snowfall for a typical winter is around 10 inches.

### [Counterfactual Ability]

*Question*: What if the Internet had been invented during the Renaissance period?

**Vicuna-7B\***: The Renaissance would have been a much different time period if the internet had been invented. The flow of information would have been much faster and more efficient, leading to a greater exchange of ideas and innovations. The world would have been connected in ways never before seen, allowing for the rapid spread of new ideas and technologies.

**LLaMA-2-7B-QFT**: The Renaissance was characterized by a renewed interest in classical learning and the arts, and the Internet would have provided a way for people to access and share this knowledge. It would have also facilitated the exchange of ideas and the development of new technologies, leading to even greater advancements in the fields of science, medicine, and engineering.

### [Generic Ability]

*Question*: Can you explain the basics of quantum computing?

**Vicuna-7B\***: Quantum computing is a type of computing that uses quantum bits (qubits) to perform calculations, unlike traditional computers that use binary.

**LLaMA-2-7B-QFT**: Quantum computing is a type of computing that uses quantum-mechanical phenomena, such as superposition and entanglement, to perform operations on data. These phenomena allow quantum computers to perform certain calculations much faster than classical computers.

### [Knowledge Ability]

*Question*: Explain the process of natural selection and how it contributes to the evolution and adaptation of species.

**Vicuna-7B\***: Natural selection is the process by which species adapt to their environment through the survival and reproduction of individuals with favorable traits. Over time, these favorable traits become more common in the population, leading to the evolution of new species.

**LLaMA-2-7B-QFT**: Natural selection is a process where species that are better adapted to their environment tend to survive and reproduce more than those that are less adapted. This leads to the survival of the fittest and the gradual evolution of species over time.

