# OpenReview forum: "QFT: Quantized Full-parameter Tuning of LLMs with Affordable Resources"
_ICLR.cc/2024/Conference — ICLR 2024 Conference Withdrawn Submission_

### Official Review · Reviewer_4QRW · 2023-10-31

**Soundness:** 2 fair
**Presentation:** 2 fair
**Contribution:** 1 poor
**Rating:** 3
**Confidence:** 5

**Summary:**

This paper proposed a framework for memory-efficient, full-parameter fine-tuning of large language models. The framework leverage Lion optimizer and the quantization of model states. The proposed methods are evaluated on LLaMA-2 models and demonstrating comparable performance to standard floating -point fine-tuning with 21% of memory usage.

**Strengths:**

1)	Paper is reasonably well written.

2)	Achieving memory reduction for fine-tuning LLaMA-2 7B/13B models with comparable accuracy with baseline.

**Weaknesses:**

1). Limited novelty. The weight quantization method in the paper is directly borrowed from the reference (SqueezeLLM). The authors over-claimed about their contribution by stating “uncover an intriguing pattern” of weight distribution. However, this pattern has already been thoroughly explained in detail in the SqueezeLLM paper and several related publications on LLM quantization.  Utilizing the Lion optimizer to save the memory for variance hardly constitutes a novel contribution.

2). Limited evaluation. The evaluation has been restricted to LLaMA models of two smaller versions, i.e. 7B and 13B. Given that the framework employs a different optimizer, it is necessary to validate its performance across a broader spectrum of models to ensure its generalizability. In addition, larger scale models are the ones really in need of memory saving, it would be beneficial to evaluate the framework on large scale models such as LLaMA 70B.

3). Lack of converging rate analysis and comparison. The paper proposed a different (lossy) optimizer but fails to provide any theoretical or empirical results regarding its converging behavior.

**Questions:**

1). It is not clear to me what is the baseline computation precision used for comparison, FP32 or FP16/BF16? For example, in Table 1, could you clarify the precisions used for columns of weights and activations?

2). In Table 1, how is the peak memory measured?  It appears that both inside of the optimizer (algorithm 2) and during forward pass (algorithm 1), all the computations are done in FP32 with all the tensors being cast into FP32 before computation. This approach seems to demand a large peak memory.

3). The algorithm introduces several quant/dequant operations before and after each operation, most of which are element-wise and, as a result, memory-intensive. Could the author provide an assessment of the computation overhead?

4). It is not clear to me how this method can be extended to larger models that require distributed computation. For example, in cases where all-reduce operations typically result high precision weights, could you explain how this method accommodates such scenarios?

5). For dense-and-sparse quantizer, could you please provide evaluations of the memory and computation overhead? The paper does not appear to include any data on this.

6). Without master copy of weights, the weight update will usually be less efficient. Could you provide ablation studies on the impact of converging rate and accuracy with or without master copy of weights?

---

### Official Review · Reviewer_wxFR · 2023-10-31

**Soundness:** 3 good
**Presentation:** 3 good
**Contribution:** 3 good
**Rating:** 6
**Confidence:** 4

**Summary:**

This paper presents a novel Quantized Full-parameter Tuning (QFT) framework for Large Language Models (LLMs), enabling memory-efficient fine-tuning for downstream tasks while preserving performance gains. The approach utilizes Lion optimization, which is conducive to quantization, along with 8-bit integer quantization for both model states and weights. The outcomes demonstrate that this method enables complete parameter fine-tuning of the LLaMA-7B model on a single A6000 GPU.

**Strengths:**

+ The proposed framework leverages the Lion optimizer's capabilities, resulting in a notable 25% reduction in memory usage for model states.
+ The strategy of quantizing various model states and parameters according to their distributions represents an interesting method for addressing memory demands.
+ Overall writing is well-structured and easily understandable.

**Weaknesses:**

- Evaluation assesses the effectiveness of the proposed framework only in terms of memory and performance efficiency.
- To further validate the proposed framework's effectiveness, the inclusion of additional models in addition to the Vicuna models would be advantageous.

**Questions:**

1. The Lion optimizer demonstrates greater efficiency when employed with larger batch sizes and lower learning rates. It would be great if the authors can address how these specific characteristics of the Lion optimizer influence the training process within their experimental setup.
2. The paper highlights QFT's competitive outcomes concerning performance and memory efficiency. Nevertheless, for a more compelling assessment, it would be beneficial to evaluate the proposed framework's performance in terms of computational complexity, particularly focusing on time complexity. Such an analysis would assist other researchers in assessing the overall cost of fine-tuning LLMs using various frameworks.

---

### Official Review · Reviewer_vKgU · 2023-11-01

**Soundness:** 3 good
**Presentation:** 2 fair
**Contribution:** 2 fair
**Rating:** 3
**Confidence:** 3

**Summary:**

This paper targets reducing memory requirements of full-parameter finetuning of LLM. It adopts quantization and Lion optimizer in order not to store full-precision weight. This paper leverages the special characteristics of Lion optimizer which has the same magnitudes for updates of every weights.
Lion optimizer only tracks momentum, whereas Adam optimizer tracks momentum and variance for optimizer states. Therefore this paper chose Lion optimizer to reduce memory requirements of finetuning LLM. In addition, this paper analyzed the distribution of gradients, momentum and weights and found that their distribution shows distinct patterns. This paper chose uniform quantizer for gradients and momentum because their distribution is centralized and has few outliers. However, it found that the distribution of weights has outliers and these outliers affect the range of limited precision. Therefore, this paper separates these outliers from the centralized values and uses a sparse matrix to represent the outliers with floating-point format. Consequently, this paper showed that its method reduces the memory usage of finetuning LLaMa-2-7b model to 21%.

**Strengths:**

This paper contributes to democratizing full-parameter finetuning of LLM. This is an important problem.
This paper showed the effectiveness of quantization with Lion optimizer to reduce memory requirements of finetuning LLM.
The LLM model quantized by the proposed methods shows comparable performance in various benchmarks compared to the baseline which uses full-precision finetuning.
The proposed gradient flow and parameter update algorithm is well-compatible with existing deep learning frameworks.

**Weaknesses:**

Overall, the paper is well written and easy to follow. However, I do not see enough innovation from this paper.
Indeed, democratizing the LLM fine-tuning is an important problem, and reducing the memory usage is one important step. The problem is that the paper seems to quantize the optimizer states, without solving much technical challenges.
The authors seem to rely on the Lion optimizer, without much reason. I understand that it only stores the momentum, but there are other optimizers that stores the same number of parameters, and there is not much connection found between the optimizer and the fine-tuning problem.

Being tied to a single optimizer is in fact not just ad-hoc but also limits the applicability of the work. Optimizers could be changed for multiple reasons, and without justification or lessons from using the specific (Lion) optimizer, users would find it difficult to generalize

I believe an indirect competitor of this work would be ZeRO-offload or ZeRO-infinity, which use host memory or storage devices to train LLM models. They will surely be slow, but they allow fine-tuning without compromising the accuracy. There should be some comparison (qualitatively and quantitatively) with those, and possibly with other fine-tuning schemes.

Using different quantizers for outliers is a widely used scheme and not new. LLM.int8() found that there are outliers in feature matrices of large transformers and used higher precision for these outliers.

**Questions:**

- How would this generalize outside the Lion optimizer?

- How would this work compare to other work along the line of democratizing LLM training?

---

### Official Review · Reviewer_FtCe · 2023-11-02

**Soundness:** 2 fair
**Presentation:** 3 good
**Contribution:** 3 good
**Rating:** 5
**Confidence:** 4

**Summary:**

The paper proposes a framework called QFT to improve the memory efficiency of LLM finetuning with minimal impact on performance. The key idea is to reduce memory requirements through low-precision quantized storage of parameters, and on-the-fly dequantization for high-precision computation to preserve performance. QFT further incorporates the Lion optimizer, which provides 25% reduction memory requirements compared to Adam. The paper provides some analysis to motivate the approach to quantizing individual model states, i.e., weights, gradients, and momentum. The evaluation results using LLAMA-2 finetuning show the benefits of the QFT on two dimensions: (i) memory efficiency compared to full precision and quantized baselines, and (ii) task performance compared to pre-trained model and full-precision finetuning.

**Strengths:**

The paper tackles an important and timely problem of how to achieve both high efficiency and performance in model finetuning.

The approach of quantized storage and full-precision computation is intuitive and promising.

The paper is very well written, organized, and easy to understand.

**Weaknesses:**

The main concern is that the draft does not demonstrate generality of QFT. I am fairly convinced of the effectiveness of QFT for LLAMA2 finetuning, but not much beyond that. The reason is that the analysis (3.2) and evaluation only studied LLAMA2 7B & 13B. Some specific questions/suggestions on this are the following:

1. It seems that QFT should be composable with Adam and bitsnbytes, and so I feel that Tables 1 & 2 would be greatly improved by adding QFT-Adam, QFT-Adam-mixed, and QFT-bitsnbytes. Is there a fundamental obstacle to this?

2. Does Figure 2 generalize to other model architectures (e.g., GPT), datasets, optimizers (e.g, Adam), etc?

Another weakness is that adopting Lion optimizer seems a weak contribution claim, since this draft does not expand knowledge of in a substantial non-obvious way.

Also, I don't think Algorithm 2 is correct for scenarios that utilize global gradient information such as gradient norms and gradient clipping.

**Questions:**

See weakness